

# Hand port-site infection after hand-assisted laparoscopic donor nephrectomy for living-donor kidney transplantation: a retrospective cohort study

Takahisa Hiramitsu[1], Toshihide Tomosugi[1], Kenta Futamura[1], Manabu Okada[1], Norihiko Goto[1], Toshihiro Ichimori[1], Shunji Narumi[1], Kazuharu Uchida[2] and Yoshihiko Watarai[1]

[1] Department of Transplant and Endocrine Surgery, Japanese Red Cross Aichi Medical Center Nagoya Daini Hospital, Nagoya, Aichi, Japan
[2] Department of Renal Transplant Surgery, Masuko Memorial Hospital, Nagoya, Aichi, Japan

Corresponding author
Takahisa Hiramitsu,
thiramitsu@outlook.jp

## ABSTRACT

**Background.** Hand-assisted laparoscopic donor nephrectomy (HALDN) is widely performed to minimize burden on living kidney donors. However, hand port-site infections after HALDN may occur. This study aimed to assess the impact of donor characteristics including preoperative comorbidities and operative factors on hand port-site infection after HALDN.

**Methods.** In this single-center, retrospective cohort study, 1,260 consecutive HALDNs for living-donor kidney transplantation performed between January 2008 and December 2021 were evaluated. All living donors met the living kidney donor guidelines in Japan. Hand port-site infections were identified in 88 HALDN cases (7.0%). To investigate risk factors for hand port-site infection, donor characteristics including preoperative comorbidities such as hypertension, glucose intolerance, dyslipidemia, obesity, and operative factors such as operative duration, blood loss, preoperative antibiotic prophylaxis, and prophylactic subcutaneous suction drain placement at the hand port-site were analyzed using logistic regression analysis.

**Results.** In the multivariate analysis, significant differences were identified regarding sex ($P = 0.021$; odds ratio [OR], 1.971; 95% confidence interval [CI], 1.108–3.507), preoperative antibiotic prophylaxis ($P < 0.001$; OR, 0.037; 95% CI [0.011–0.127]), and prophylactic subcutaneous suction drain placement at the hand port-site ($P = 0.041$; OR, 2.005; 95% CI [1.029–3.907]). However, a significant difference was not identified regarding glucose intolerance ($P = 0.572$; OR, 1.148; 95% CI [0.711–1.856]). Preoperative comorbidities may not cause hand port-site infections within the donors who meet the living kidney donor guidelines. Preoperative antibiotic prophylaxis is crucial in preventing hand port-site infection, whereas prophylactic subcutaneous suction drain placement may increase the risk of hand port-site infection.

prophylaxis, Rophylactic subcutaneous suction drain placement, Glucose intolerance, Smoking history

## INTRODUCTION

Because of an extreme organ shortage for transplantation, the number of living-donor kidney transplantations has been steadily increasing (*Organ Procurement and Transplantation Network*). To reduce the burden on living kidney donors, hand-assisted laparoscopic donor nephrectomy (HALDN) has been developed (*Ratner et al., 1995*; *Wolf, Tchetgen & Merion, 1998*). Numerous reports have demonstrated the safety and efficacy of HALDN in donors and recipients (*Buell et al., 2002*; *Nakajima et al., 2012*; *Rajab & Pelletier, 2015*; *Kumar et al., 2018*; *Hiramitsu et al., 2021*). However, superficial surgical site infection after HALDN is observed in 4.8%–20.7% of cases (*Nakajima et al., 2012*; *Ahmed et al., 2020*; *Hiramitsu et al., 2021*). A recent report from the United Kingdom demonstrated the efficacy of preoperative antibiotic prophylaxis in preventing superficial surgical site infection after HALDN (*Ahmed et al., 2020*). Thus, the previous HALDN guidelines,which suggest that preoperative antibiotic prophylaxis for HALDN is unnecessary because the operation is performed aseptically, have now come into question (*National Institute for Health and Care Excellence, 2017*). However, the impact of operative factors, such as prophylactic subcutaneous drain placement, and preoperative comorbidities, such as hypertension, glucose intolerance, and dyslipidemia that could cause superficial surgical site infection were not investigated in the report (*Ahmed et al., 2020*). In previous studies, the efficacy of prophylactic subcutaneous suction drain placement has been controversial; however, these studies did not focus on living-donor nephrectomy (*Panici et al., 2003*; *Hellums, Lin & Ramsey, 2007*; *Baier et al., 2010*; *Kosins et al., 2013*; *Coletta et al., 2019*). Preoperative comorbidities were identified as risk factors for surgical site infections (*Mangram et al., 1999*). However, the impact of preoperative comorbidities on surgical site infections has not been investigated in living kidney donors. This study investigated the impact of donor characteristics including preoperative comorbidities and operative factors on hand port-site infection after HALDN. Our aim to assess the impact of donor characteristics was achieved with a retrospective cohort study and statistical analysis using data from donors who underwent HALDN.

## MATERIALS & METHODS

### Study design

This retrospective cohort study was approved by the Institutional Review Board of Nagoya Daini Red Cross Hospital (Aichi, Japan) (approval number: 1068) and was conducted according to the principles of the Declaration of Helsinki. Living-donor kidney transplantation was performed according to the Declaration of Istanbul. Living kidney donors were classified into infection and no infection groups based on the occurrence of hand port-site infection. Donor characteristics including preoperative comorbidities and operative factors were analyzed using logistic regression analysis, to investigate their impact

on hand port-site infection. This study was reported in accordance with the Strengthening the Reporting of Observational Studies in Epidemiology (STROBE) guidelines.

## Participants and follow-up assessments

We recruited all consecutive living donors who underwent HALDN at our hospital between January 2008 and December 2021. A total of 1,260 HALDN cases were enrolled in this study. No donors were excluded from the analysis. All donor data were retrospectively obtained from medical records and analyzed anonymously. Thus, the need for informed consent was waived by the institutional review board.

All patients were followed up. Postoperative assessments of donors for hand port-site infection were performed after HALDN for 1 month (*Holihan et al., 2017*).

## Living donors and hand-assisted laparoscopic donor nephrectomy

Living donors were selected according to the living kidney donor guidelines in Japan (*Hiramitsu et al., 2020*). After general anesthesia induction, a single dose of prophylactic intravenous antibiotic (primarily first-generation cephalosporins) was administered for the preoperative antibiotic prophylaxis. Clindamycin was administered to living kidney donors who were allergic to specific antibiotics. The skin was prepared with povidone-iodine; for living kidney donors who were allergic to iodine, chlorhexidine was used (*Rodrigues & Simøes, 2013*). HALDN was performed as previously described (*Hiramitsu et al., 2021*). A 7.5-cm mid-epigastric incision was made for the hand port device in the left HALDN and a 7.5-cm lower-quadrant pararectal incision was made for the hand port device in the right HALDN (*Hiramitsu et al., 2021*). A prophylactic closed subcutaneous suction drain was placed at the hand port-site. The drain was removed at 1–5 days postoperatively (*Allegranzi et al., 2016*). Between January 2008 and June 2015, neither a single dose of prophylactic intravenous antibiotic was administered nor a prophylactic closed subcutaneous suction drain was used routinely. Between July 2015 and May 2016, the prophylactic closed subcutaneous suction drain without the single dose of prophylactic intravenous antibiotic was routinely used. Between June 2016 and December 2020, both the single dose of prophylactic intravenous antibiotic and prophylactic closed subcutaneous suction drain were routinely used. Since January 2021, the single dose of prophylactic intravenous antibiotic without prophylactic closed subcutaneous suction drain has been used routinely. Donors routinely left the hospital 7 days after donor nephrectomy, if they did not have any complications.

## Definition of hand port-site infection

Hand port-site infection was diagnosed based on the Centers for Disease Control and Prevention criteria as follows (*Horan et al., 1992*): purulent drainage from the wound; infection confirmed by microbiological testing of the wound; and at least two of the following symptoms: fever; pain, swelling, redness, and heat at the wound; and wound dehiscence.

## Definition of donor comorbidities

Hypertension was defined as blood pressure >140/90 mmHg or treatment with antihypertensives. Dyslipidemia was defined as low-density lipoprotein cholesterol level
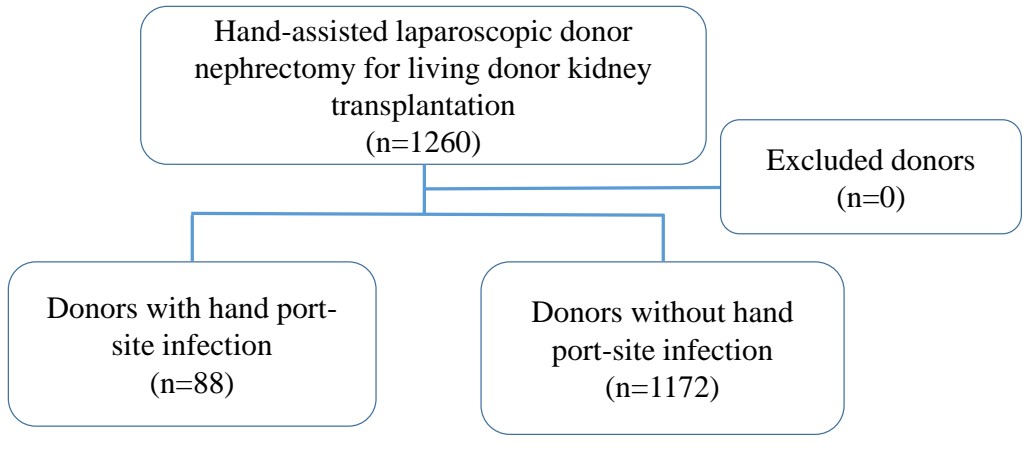

**Figure 1** Flow chart of patient inclusion.

>140 mg/dl, triglyceride level >150 mg/dl, and high-density lipoprotein cholesterol level <40 mg/dl or treatment with lipid-lowering agents. Glucose intolerance was defined as impaired fasting glycemia, impaired glucose tolerance, or diabetes mellitus.

## Statistical analysis

Statistical analyses of donor characteristics were performed using the Mann–Whitney U test for continuous variables and the Chi-square test or Fisher's exact test for categorical variables. Logistic regression analysis was used to determine the factors contributing to hand port-site infection. The recruiting period for living donors was extensive; hence, to minimize the impact of a long recruiting period, we divided the study population into two groups: first operation period (January 2008–December 2015) and second operation period (January 2016–December 2021). A $P$ value <0.05 was considered statistical significant. SPSS® software, version 23.0 for Windows (IBM Corporation, Armonk, NY, USA) was used for statistical analyses.

## RESULTS

### Study population

A total of 1,260 HALDNs for living-donor kidney transplantation were performed at our hospital during the study period. All donors ($n = 1,260$) were included in this study and were followed up after HALDN for 1 month to assess the incidence of hand port-site infections. Hand port-site infection was identified in 88 donors (infection group). The remaining 1,172 donors did not experience hand port-site infection (no infection group) (Fig. 1).

### Donor characteristics

Table 1 presents the characteristics of the donors, including the preoperative characteristics and operation results. Significant differences in sex ($P < 0.001$), smoking history ($P = 0.002$), glucose intolerance ($P = 0.003$), glycated hemoglobin (HbA1c) ($P = 0.023$),

**Table 1 Donor characteristics.**

| | No infection group $n = 1{,}172$ | Infection group $n = 88$ | P-value | Odds ratio | 95% CI Lower limit | 95% CI Upper limit |
|---|---|---|---|---|---|---|
| Age (years, IQR) | 60.0 (56.0–67.0) | 65.0 (63.0–67.5) | 0.804[***] | | | |
| Sex (male, %) | 409 | 49 | **<0.001**[*] | 2.344 | 1.513 | 3.630 |
| Smoking history (%) | 507 | 53 | **0.002**[*] | 1.986 | 1.276 | 3.091 |
| Hypertension (%) | 347 | 23 | 0.490[*] | 0.841 | 0.514 | 1.376 |
| Dyslipidemia (%) | 688 | 53 | 0.779[*] | 1.065 | 0.684 | 1.658 |
| Glucose intolerance (%) | 290 | 35 | **0.003**[*] | 2.008 | 1.284 | 3.141 |
| Obesity (body mass index $\geq 30$ kg/m$^2$, %) | 8 | 1 | 0.480[**] | 1.672 | 0.207 | 13.525 |
| Preoperative systolic blood pressure (mmHg, IQR) | 130.0 (121.0–140.0) | 118.5 (113.5–131.0) | 0.922[***] | | | |
| Preoperative diastolic blood pressure (mmHg, IQR) | 75.0 (65.0–86.0) | 72.0 (70.0–80.5) | 0.375[***] | | | |
| Preoperative total cholesterol (mg/dl, IQR) | 206.0 (193.0–231.0) | 251.0 (205.5–267.0) | 0.157[***] | | | |
| Preoperative triglyceride (mg/dl, IQR) | 171.0 (119.0–198.0) | 119.5 (87.5–152.0) | 0.111[***] | | | |
| Preoperative low-density lipoprotein cholesterol (mg/dl, IQR) | 120.0 (107.0–145.0) | 156.0 (132.0–170.5) | 0.422[***] | | | |
| Preoperative fasting glucose level (mg/dl, IQR) | 98.0 (95.0–104.0) | 97.0 (86.5–106.5) | 0.341[***] | | | |
| 75-g oral glucose tolerance test results (blood glucose level 2 h after glucose administration, mg/dl, IQR) | 124.0 (112.0–162.0) | 122.0 (105.0–141.5) | 0.303[***] | | | |
| HbA1c (%, IQR) | 5.7 (5.6–5.7) | 5.7 (5.6–5.8) | **0.023**[***] | | | |
| Body mass index (kg/m$^2$, IQR) | 25.1 (20.4–26.6) | 24.3 (22.2–24.8) | 0.179[***] | | | |
| Kidney laterality (left, %) | 1086 | 84 | 0.397[*] | 1.663 | 0.596 | 4.643 |
| Preoperative eGFR (ml/min/1.73 m$^2$, IQR) | 77.8 (66.8–83.8) | 67.6 (62.3–71.1) | 0.703[***] | | | |

**Table 1** (*continued*)

| | | No infection group $n = 1{,}172$ | Infection group $n = 88$ | P-value | Odds ratio | 95% CI Lower limit | Upper limit |
|---|---|---|---|---|---|---|---|
| Preoperative urine albumin/Cr ratio (mg/gCr, IQR) | | 9.86 (5.68–12.35) | 6.37 (0.27–13.99) | 0.050[***] | | | |
| Preoperative antibiotic prophylaxis (%) | | 643 | 4 | **<0.001**[**] | 0.039 | 0.014 | 0.108 |
| Prophylactic subcutaneous suction drain placement at hand port site (%) | | 555 | 21 | **<0.001**[*] | 0.348 | 0.211 | 0.576 |
| Countermeasure for hand port-site infection | No preoperative antibiotic prophylaxis or prophylactic subcutaneous suction drain placement at hand-port site | 452 (38.6) | 68 (77.2) | | | | |
| | Preoperative antibiotic prophylaxis alone | 162 (13.8) | 2 (2.3) | | | | |
| | Prophylactic subcutaneous suction drain placement at hand-port site alone | 73 (6.2) | 20 (22.7) | **<0.001**[*] | | | |
| | Preoperative antibiotic prophylaxis and prophylactic subcutaneous suction drain placement at hand-port site | 481 (38.2) | 2 (0.2) | | | | |
| Operation period | First operation period (January 2008–December 2015, %) | 554 (44.0) | 78 (6.2) | | | | |
| | Second operation period (January 2016–December 2021, %) | 618 (49.0) | 10 (0.8) | **<0.001**[**] | 0.115 | 0.059 | 0.224 |
| Follow-up period <1 month (%) | | 0 | 0 | n.c. | | | |

**Notes.**

CI, confidence interval; IQR, interquartile range; HbA1c, hemoglobin A1c; eGFR, estimated glomerular filtration rate; n.c., not calculation.

Boldface indicates statistically significant results.

[*]Chi-square test.

[**]Fisher's exact test.

[***]Mann–Whitney $U$ test.

preoperative antibiotic prophylaxis ($P < 0.001$), prophylactic subcutaneous suction drain placement at the hand port-site ($P < 0.001$), subgroups classified by preoperative antibiotic prophylaxis and prophylactic subcutaneous suction drain placement at the hand port-site ($P < 0.001$), and operation period ($P < 0.001$) were observed between the infection and no infection groups.

### Donor surgical outcomes
A significant difference was identified in operative duration between the infection and no infection groups ($P = 0.002$) (Table 2).

### Diagnosis of hand port-site infection
Purulent drainage from the wound was identified in 42 (47.7%) living kidney donors. Infection confirmed by microbiological testing from the wound was identified in 15 (17.0%) donors. At least two of the following symptoms: fever, pain, swelling, redness, and heat at the wound and wound dehiscence were identified in 88 (100%) donors.

### Treatment of hand port-site infection
Donors with hand port-site infection were mainly treated with drainage and lavage (78.4%). For the treatment of hand port-site infection, further admission was not identified. No significant difference was noted in prolonged hospital stay between the infection and no infection groups (Table 2).

### Hand port-site hernia
Hand port-site hernia was significantly more identified in the infection group than in the no infection group ($P = 0.028$; odds ratio [OR], 5.874; 95% CI [1.492–23.123]; Table 2)

### Risk of hand port-site infection
Tables 3 and 4 present the results of the logistic regression analysis which was performed to identify the risk factors for hand port-site infection. In the univariate analysis, significant differences in sex ($P < 0.001$; OR, 2.344; 95% confidence interval CI [1.513–3.630], smoking history ($P = 0.002$; OR, 1.986; 95% CI [1.276–3.091]), glucose intolerance ($P = 0.002$; OR, 2.008; 95% CI [1.284–3.141]), preoperative antibiotic prophylaxis ($P < 0.001$; OR, 0.039; 95% CI [0.014–0.108]), and prophylactic subcutaneous suction drain placement at the hand port-site ($P < 0.001$; OR, 0.348; 95% CI [0.211–0.576]), and first operation period ($P < 0.001$; OR, 0.115; 95% CI [0.059–0.224]) were identified. In the multivariate analysis, significant differences in sex ($P = 0.021$; OR, 1.971; 95% CI [1.108–3.507]), preoperative antibiotic prophylaxis ($P < 0.001$; OR, 0.037; 95% CI [0.011–0.127]), and prophylactic subcutaneous suction drain placement at the hand port-site ($P = 0.041$; OR, 2.005; 95% CI [1.029–3.907]) were identified. However, a significant difference was not identified regarding smoking history ($P = 0.120$; OR, 1.572; 95% CI [0.888–2.784]), glucose intolerance ($P = 0.572$; OR, 1.148; 95% CI [0.711–1.856]), and first operation period ($P = 0.459$; OR, 0.698; 95% CI [0.269–1.810]).

**Table 2  Donor surgical outcomes.**

| | | No infection group | Infection group | P-value | Odds ratio | 95% CI | |
|---|---|---|---|---|---|---|---|
| | | n = 1,172 | n = 88 | | | Lower limit | Upper limit |
| Operative duration (min, IQR) | | 207.0 (185.0–238.0) | 262.5 (200.5–302.0) | **0.002**[***] | | | |
| Blood loss (min, IQR) | | 32.0 (10.0–40.0) | 22.5 (5.0–47.5) | 0.758[***] | | | |
| Perioperative adverse events (%) | | 20 (1.7) | 0 | 0.391[**] | | | |
| Open conversion (%) | | 3 (0.3) | 0 | >0.999[**] | | | |
| Intraoperative bleeding (%) | | 2 (0.2) | 0 | >0.999[**] | | | |
| Renal artery injury (%) | | 1 (0.1) | 0 | >0.999[**] | | | |
| Postoperative bleeding requiring reoperation (%) | | 2 (0.2) | 0 | >0.999[**] | | | |
| Intestinal injury (%) | | 1 (0.1) | 0 | >0.999[**] | | | |
| Pneumonia (%) | | 3 (0.3) | 0 | >0.999[**] | | | |
| Urinary tract infection (%) | | 10 (0.9) | 0 | >0.999[**] | | | |
| Small bowel obstruction (%) | | 1 (0.1) | 0 | >0.999[**] | | | |
| Duration of hospital stay (days, IQR) | | 7.3 (0.9) | 7.5 (1.1) | 0.306[***] | | | |
| Readmission for the treatment of hand-port site infection (%) | | 0 | 0 | n.c. | | | |
| Treatment for the hand-port site infection | Antibiotics (%) | 0 | 1 (1.1) | | | | |
| | Drainage and lavage (%) | 0 | 69 (78.4) | | | | |
| | Drainage, lavage, and antibiotics (%) | 0 | 16 (18.2) | **<0.001**[*] | | | |
| | Drainage, lavage, antibiotics, and debridment (%) | 0 | 2 (2.3) | | | | |
| Hand-port site hernia (%) | | 7 (0.6) | 3 (3.4) | **0.028**[**] | 5.874 | 1.492 | 23.123 |

**Notes.**
CI, confidence interval; IQR, interquartile range; n.c., not calculation.
Boldface indicates statistically significant results.
[*]Chi-square test.
[**]Fisher's exact test.
[***]Mann–Whitney $U$ test.

## DISCUSSION

This study demonstrated the efficacy of preoperative antibiotic prophylaxis for the prevention of hand port-site infections and showed that prophylactic subcutaneous suction drain placement at the hand port-site may increase the risk of developing hand

**Table 3 Univariate logistic regression analysis for the risk of port-site infection.**

| | B | Univariate analysis | | | |
| --- | --- | --- | --- | --- | --- |
| | | *P*-value | Odds ratio | 95% CI | |
| | | | | Lower limit | Upper limit |
| Age (years) | 0.007 | 0.546 | 1.007 | 0.985 | 1.029 |
| Sex (male *vs.* female) | 0.852 | **<0.001** | 2.344 | 1.513 | 3.630 |
| Smoking history (*vs.* no smoking history) | 0.686 | **0.002** | 1.986 | 1.276 | 3.091 |
| Hypertension (*vs.* no hypertension) | −0.173 | 0.491 | 0.841 | 0.514 | 1.376 |
| Dyslipidemia (*vs.* no dyslipidemia) | 0.063 | 0.779 | 1.065 | 0.684 | 1.658 |
| Glucose intolerance (*vs.* no glucose intolerance) | 0.697 | **0.002** | 2.008 | 1.284 | 3.141 |
| Obesity (body mass index $\geq$30 kg/m$^2$) | 0.514 | 0.630 | 1.672 | 0.207 | 13.525 |
| Body mass index (kg/m$^2$) | 0.048 | 0.200 | 1.050 | 0.975 | 1.130 |
| Kidney laterality (left *vs.* right) | 0.509 | 0.332 | 1.663 | 0.596 | 4.643 |
| Preoperative eGFR (ml/min/1.73 m$^2$) | −0.004 | 0.646 | 0.996 | 0.979 | 1.013 |
| Preoperative urine albumin/Cr ratio (mg/gCr) | −0.021 | 0.203 | 0.980 | 0.949 | 1.011 |
| Preoperative antibiotic prophylaxis (*vs.* no preoperative antibiotic prophylaxis) | −3.240 | **<0.001** | 0.039 | 0.014 | 0.108 |
| Prophylactic subcutaneous suction drain placement at hand port site (*vs.* no prophylactic subcutaneous suction drain placement at hand port site) | −1.054 | **<0.001** | 0.348 | 0.211 | 0.576 |
| Operative duration (min) | 0.001 | 0.169 | 1.001 | 1.000 | 1.003 |
| Blood loss (min) | 0.001 | 0.632 | 1.001 | 0.998 | 1.003 |
| Perioperative adverse events (*vs.* no perioperative adverse events) | −18.631 | 0.998 | <0.001 | 0 | |
| First operation period (*vs.* second operation period) | −2.163 | **<0.001** | 0.115 | 0.059 | 0.224 |

Notes.
CI, confidence interval; HbA1c, hemoglobin A1c; eGFR, estimated glomerular filtration rate.
Boldface indicates statistically significant results.

**Table 4 Multivariate logistic regression analysis for the risk of port-site infection.**

| | B | Multivariate analysis | | | |
| --- | --- | --- | --- | --- | --- |
| | | *P*-value | Odds ratio | 95% CI | |
| | | | | Lower limit | Upper limit |
| Sex (male *vs.* female) | 0.679 | **0.021** | 1.971 | 1.108 | 3.507 |
| Smoking history (*vs.* no smoking history) | 0.453 | 0.120 | 1.572 | 0.888 | 2.784 |
| Glucose intolerance (*vs.* no glucose intolerance) | 0.138 | 0.572 | 1.148 | 0.711 | 1.856 |
| Preoperative antibiotic prophylaxis (*vs.* no preoperative antibiotic prophylaxis) | −3.294 | **<0.001** | 0.037 | 0.011 | 0.127 |
| Prophylactic subcutaneous suction drain placement at hand port site (*vs.* no prophylactic subcutaneous suction drain placement at hand port site) | 0.696 | **0.041** | 2.005 | 1.029 | 3.907 |
| First operation period (*vs.* second operation period) | −0.360 | 0.459 | 0.698 | 0.269 | 1.810 |

Notes.
CI, confidence interval; HbA1c, hemoglobin A1c; eGFR, estimated glomerular filtration rate.
Boldface indicates statistically significant results.

port-site infections. The preoperative comorbidities were not found to be associated with the risk of developing hand port-site infections.

The incidence of hand port-site infection in this study was 7.0%, which is similar to the results of previous studies (*Nakajima et al., 2012*; *Ahmed et al., 2020*; *Favi et al., 2020*; *Hiramitsu et al., 2021*; *Vaz et al., 2022*). All donors showed at least two symptoms for the diagnosis of hand port-site infection. However, only 17.0% of donors were cases of infection confirmed by microbiological testing from the wound.

We investigated hand port-site infection further by analyzing donor factors (including comorbidities, such as hypertension, dyslipidemia, glucose intolerance, and obesity) and operative factors (including operative duration, blood loss, preoperative antibiotic prophylaxis, and prophylactic subcutaneous suction drain placement at the hand port-site). Glucose intolerance was more frequent in the infection group than in the no infection group. No significant differences in the preoperative fasting glucose level or blood glucose level at 2 h after glucose administration in the 75-g oral glucose tolerance test were identified except for HbA1c. According to the living kidney donor guidelines in Japan, living donor candidates with glucose intolerance who meet the marginal criteria are allowed to be considered as a candidate; the marginal criteria include well-managed usage of oral anti-diabetic agents and an HbA1c ≤6.5% and a urine albumin/Cr ratio <30 mg/g Cr. However, candidates receiving insulin were ineligible for donation (*Hiramitsu et al., 2020*). Since compliance to these guidelines implied that donors with severe glucose intolerance were not included in this study, our multivariate analysis may conclude that glucose intolerance may not be a risk factor for hand port-site infection. Other comorbidities including hypertension, dyslipidemia, and obesity were similar in both groups, which could be because the living kidney donors were selected according to the living kidney donor guidelines in Japan (*Hiramitsu et al., 2020*). Additionally, according to these guidelines, a blood pressure <140/90 mmHg and BMI ≤30 kg/m$^2$ (patients with high BMI were encouraged to reduce weight to achieve BMI ≤25 kg/m$^2$) were recommended. Living donor candidates who did not meet the blood pressure criteria could be a living donor if they met the marginal criteria, which included candidates with hypertension who had maintained their blood pressure at ≤130/80 mmHg and had an albumin/Cr ratio of <30 mg/g Cr. These candidates were ruled out with the following conditions: an organic disorder due to hypertension such as myocardial hypertrophy, changes to the fundus of the eye, and severe aortic calcification. Living donor candidates with a BMI ≤32 kg/m$^2$ were allowed to be living kidney donors, and they were encouraged to reduce weight to achieve a BMI ≤25 kg/m$^2$. Overall, compliance to these criteria implied that donors with severe dyslipidemia, severe hypertension, and obesity were not included in this study.

Although smoking history was more frequent in the infection group than in the no infection group, the living kidney donors were strictly instructed to stop smoking before the operation to prevent other complications and to maintain the function of the remaining kidney. In this study, current smokers were not included.

The univariate analysis showed that sex, smoking history, and glucose intolerance significantly increase the risk of hand port-site infection. In previous reports, smoking history and glucose intolerance were identified as risk factors for superficial surgical site

infection (*Mangram et al., 1999*; *Sørensen, 2012*; *Martin et al., 2016*; *Anderson et al., 2017*). Based on the multivariate analysis, male sex and prophylactic subcutaneous suction drain placement at the hand port-site significantly increased the risk of hand port-site infection. Smoking history and glucose intolerance may not cause hand port-site infection because living kidney donors were well examined and prepared before the operation according to the living kidney donor guidelines in Japan (*Hiramitsu et al., 2020*). In our study, only preoperative antibiotic prophylaxis reduced the risk of hand port-site infection, which is consistent with the finding of a previous prospective study in the United Kingdom (*Ahmed et al., 2020*). However, the previous study did not investigate the effect of prophylactic subcutaneous suction drain placement in living-donor nephrectomy. Moreover, the efficacy of prophylactic subcutaneous suction drain placement in other surgeries is controversial (*Panici et al., 2003*; *Hellums, Lin & Ramsey, 2007*; *Baier et al., 2010*; *Kosins et al., 2013*; *Coletta et al., 2019*). A previous study has concluded that prophylactic subcutaneous suction drain placement could cause surgical site infection and noted that the prophylactic subcutaneous suction drain is an entry point for pathogens (*Kosins et al., 2013*). This finding was in agreement with the results of our study.

After the identification of hand port-site infection, donors were mainly treated with drainage and lavage (78.4%). For the treatment of hand port-site infection, further admission was not identified in this study. The period of hospital stay was similar between the infection and no infection group. Hand port-site hernia was significantly more identified in the infection group than in the no infection group, as previously reported (*Favi et al., 2020*). As hand port-site infection required additional treatment and had other disadvantages to donors, preoperative antibiotic prophylaxis is critical. For efficient preoperative prophylactic antibiotic selection, preoperative microbiology screening may be appreciated.

The major limitation of this study was its retrospective design, long recruiting period for living donors with multiple changes of preoperative management, the lack of characterization of the pathogens involved, and limited follow-up data. Therefore, large prospective randomized controlled studies, including the preoperative comorbidities of living donors, are warranted. To minimize the impact of a long recruiting period, we divided the study population into two groups, first operation period (January 2008–December 2015) and second operation period (January 2016–December 2021), and performed analyses using the logistic regression models. Nevertheless, this study demonstrated the effects of preoperative donor factors, preoperative antibiotic prophylaxis, and prophylactic subcutaneous suction drain placement on the development of hand port-site infection.

## CONCLUSIONS

In conclusion, preoperative antibiotic prophylaxis may decrease the incidence of hand port-site infection, whereas prophylactic subcutaneous suction drain placement at the hand port-site may adversely increase the risk of superficial surgical site infection. Moreover, preoperative comorbidities may not cause hand port-site infection in donors who met the living kidney donor guidelines. To prevent these infections, preoperative antibiotic prophylaxis may be required.

## ACKNOWLEDGEMENTS

We would like to thank Editage for English language editing.

### Funding

The authors received no funding for this work.

### Competing Interests

The authors declare there are no competing interests.

### Author Contributions

- Takahisa Hiramitsu conceived and designed the experiments, prepared figures and/or tables, authored or reviewed drafts of the article, and approved the final draft.
- Toshihide Tomosugi performed the experiments, prepared figures and/or tables, and approved the final draft.
- Kenta Futamura analyzed the data, prepared figures and/or tables, and approved the final draft.
- Manabu Okada analyzed the data, authored or reviewed drafts of the article, and approved the final draft.
- Norihiko Goto analyzed the data, authored or reviewed drafts of the article, and approved the final draft.
- Toshihiro Ichimori performed the experiments, authored or reviewed drafts of the article, and approved the final draft.
- Shunji Narumi performed the experiments, authored or reviewed drafts of the article, and approved the final draft.
- Kazuharu Uchida performed the experiments, authored or reviewed drafts of the article, and approved the final draft.
- Yoshihiko Watarai performed the experiments, prepared figures and/or tables, authored or reviewed drafts of the article, and approved the final draft.

### Human Ethics

The following information was supplied relating to ethical approvals (*i.e.*, approving body and any reference numbers):

The Institutional Review Board of Nagoya Daini Red Cross Hospital approved the study (1068).

### Data Availability

The data is available at Figshare: Hiramitsu, Takahisa (2022): Hand port site infection. figshare. Dataset. https://doi.org/10.6084/m9.figshare.20055269.v2.

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
