# Peer review of "Hand port-site infection after hand-assisted laparoscopic donor nephrectomy for living-donor kidney transplantation: a retrospective cohort study"

_PeerJ, doi:10.7717/peerj.14215_

## Round 0.1 · original submission · Major Revisions

Dear Authors, thank you for your submission to PeerJ. You are requested to please consider the comments of both referees. There is a need to address the limitations of the manuscripts, along with methodological disparities.

·

Basic reporting

Since the observation period spans over 13 years, the authors should examine if the period could be a factor or not. For example, compare the incidence between the first and second half observation periods. Furthermore, this factor should be put into the logistic regression model.

Abstract: After the phrase, 'Hand port-site infections were identified in 88 HALDN cases.' please specify the actual incident rate, which is 88/1260 = nearly 7%.

Table 1. Surgical outcomes are not Donor Characteristics. Please stick to the Title's indication. I recommend you should cut this Table into two retaining the first one as pure Donor Characteristics. A new Table can be called Surgical Outcomes, which are essential study findings.

The following two studies should be cited and assimilated into your main text:
#1. Vaz O, et al. Laterality in laparoscopic hand assisted donor nephrectomy - Does it matter anymore? Outcomes of a large retrospective series. Surgeon. 2021 Nov 26:S1479-666X(21)00163-3.
#2. Evaldo Favi and colleagues. Outcomes and surgical complications following living-donor renal transplantation using kidneys retrieved with trans-peritoneal or retro-peritoneal hand-assisted laparoscopic nephrectomy. Clin Transplant 2020 Dec;34(12):e14113.

Experimental design

This observation study over 13 years creates medical innovation and period-related confounding factors. As I have mentioned in the Basic Reporting, the authors should include the period factor in the logistic regression model.

Validity of the findings

According to the literature, the authors have reported their infection rate from their institution, which is a not-bad outcome; the authors should then compare their data with the national one.

Additional comments

No further comments.

Reviewer 2 ·

Basic reporting

no comment

Experimental design

The authors report on a very long streak (2008-2021) of HALDN. Over time, they have changed their policy regarding prophylactic antibiotic use and subcutaneous drainage placement. As such, you may count several different groups of patients: 1) antibiotic - and drainage -; 2) Antibiotic + and drainage -; 3) antibiotic - and drainage +; 4) antibiotic + and drainage +. For clarity purposes, these subgroups should be compared, aiming to show possible differences in surgical site infection rates. Also, reporting on the number of patients receiving different peri-operative management would help the readers ruling out the possiility of selection bias.

Validity of the findings

This study has several limitations other than the retrospective nature. The fact that donors management has changed multiple times over the observation period is extremely relevant as much as the lack of characterization of the pathogens involved and the limited follow-up data. Indeed, as reported in Clin Transplant. 2020 Dec;34(12):e14113, surgical site infection may be associated or promote the development of hand-port incisional hernia, thus adding further complications. Also, it would be relevant to know how patients with surgical site infection were treated and if they had prolonged hospitalization, or further hospital admissions.

Additional comments

A brief discussion on the possible benefits of pre-operative microbiology screening would be appreciated.

---

## Round 0.2 · accepted · Accept

Thanks for considering the suggestions and revising the draft accordingly.

·

Basic reporting

no comment.

Experimental design

no comment.

Validity of the findings

no comment.

Additional comments

I think this revised manuscript has attained publishable quality, and I am happy to recommend acceptance for publication in PeerJ.

Reviewer 2 ·

Basic reporting

No comment

Experimental design

No comment

Validity of the findings

No comment

Additional comments

The manuscript has been much improved. As such, it can be published without further revision.